# Can One Series of Self-Organized Nanoripples Guide Another Series of Self-Organized Nanoripples during Ion Bombardment: From the Perspective of Power Spectral Density Entropy?

**DOI:** 10.3390/e25010170

**Published:** 2023-01-14

**Authors:** Hengbo Li, Jinyu Li, Gaoyuan Yang, Ying Liu, Frank Frost, Yilin Hong

**Affiliations:** 1National Synchrotron Radiation Laboratory, University of Science and Technology of China, Hezuohua South Road 42, Hefei 230029, China; 2Leibniz Institute of Surface Engineering (IOM), Permoserstraße 15, 04318 Leipzig, Germany

**Keywords:** self-organization, self-assembly, guided self-organization, ion bombardment, entropy, PSD entropy, nanostructures, nanoripples, degree of ordering

## Abstract

Ion bombardment (IB) is a promising nanofabrication tool for self-organized nanostructures. When ions bombard a nominally flat solid surface, self-organized nanoripples can be induced on the irradiated target surface, which are called intrinsic nanoripples of the target material. The degree of ordering of nanoripples is an outstanding issue to be overcome, similar to other self-organization methods. In this study, the IB-induced nanoripples on bilayer systems with enhanced quality are revisited from the perspective of guided self-organization. First, power spectral density (PSD) entropy is introduced to evaluate the degree of ordering of the irradiated nanoripples, which is calculated based on the PSD curve of an atomic force microscopy image (i.e., the Fourier transform of the surface height. The PSD entropy can characterize the degree of ordering of nanoripples). The lower the PSD entropy of the nanoripples is, the higher the degree of ordering of the nanoripples. Second, to deepen the understanding of the enhanced quality of nanoripples on bilayer systems, the temporal evolution of the nanoripples on the photoresist (PR)/antireflection coating (ARC) and Au/ARC bilayer systems are compared with those of single PR and ARC layers. Finally, we demonstrate that a series of intrinsic IB-induced nanoripples on the top layer may act as a kind of self-organized template to guide the development of another series of latent IB-induced nanoripples on the underlying layer, aiming at improving the ripple ordering. The template with a self-organized nanostructure may alleviate the critical requirement for periodic templates with a small period of ~100 nm. The work may also provide inspiration for guided self-organization in other fields.

## 1. Introduction

Self-organization [1,2,3,4,5] occurs in diverse fields such as physics, chemistry, and biology. To tailor and utilize self-organization, guided self-organization (GSO) has emerged as a special research field for nearly two decades [6,7,8,9,10,11]. It is still an open question as to how to guide self-organization. Due to the roughening and smoothing mechanisms during ion bombardment (IB) [12,13,14,15,16,17,18,19], a broad and uniform beam of energetic ions may induce self-organized nanostructures on an initial flat solid surface, in particular, quasiperiodic nanoripples under oblique incidence [12]. Being promising in the applications of different fields [20,21,22], IB-induced nanoripples have attracted increasing interest and are also representative self-organized nanostructures in the physical field. As self-organization emerges at multiple scales and involves multiple domains, research on GSO should also involve multiple fields and be even more complex than research on self-organization. Research on IB may provide inspiration and a reference for GSOs in other fields.

From the point of view of GSO, the guided IB corresponds to the bombardment of an initially non-flat solid surface or that of a prepatterned surface. Since the degree of ordering of nanostructures has long been an important issue for self-organized patterns, the morphological evolution of a prepatterned surface during IB has been studied to enhance the quality of IB-induced nanoripples. Naturally, prepatterns with periodic structures should be a first candidate for guiding the evolution of nanoripples during IB. However, prepatterns with periodic structures can be fabricated by focus ion beam milling or electron beam lithography [8], neither of which is compatible with the cost-effective and efficient method of IB. Hence, the use of self-organized nanoripples as a prepattern can guide the growth of nanoripples, which may alleviate the critical requirements for periodic prepatterns.

Recently, our group proposed and demonstrated the enhancement of the quality of self-organized nanoripples by the bombardment of optimized bilayers [23,24]. During the bombardment of a bilayer, the nanoripples produced on the underlying layer can be guided by the nanoripples on the top layer.

To deepen the understanding of the IB of a bilayer, in this work, referring to the Shannon information entropy for the evaluation of a system complexity, we introduce a parameter of PSD entropy to characterize the degree of ordering (i.e., the lateral periodicity of nanoripples). To the best of our knowledge, this parameter has not been reported in previous evaluations of IB-induced nanostructures. To study the effect of IB on surface topography, irradiated surfaces are generally measured by using atomic force microscopy (AFM). Extracted from AFM data, PSD, autocorrelation length [16] and defect density [19] are often used to evaluate the regularity of self-organized structures. Recently, the system correlation length (i.e., lateral correlation length) has been used to characterize the lateral ordering of structures. The system correlation length is calculated from the reciprocal of the full width at half maximum (FWHM) of the PSD peak. Regarding nanostructures without pronounced peaks in their PSD curves, it is difficult to measure their FWHM and calculate their correlation length. Thus, it is necessary to consider a parameter to characterize the degree of ordering of nanostructures without a remarkable peak in PSD curves. Calculated based on the PSD curves from AFM data, PSD entropy is a value used to evaluate the degree of ordering of nanoripples. In addition, the PSD entropy of the nanoripples is compared with the PSD curves and root-mean-square (rms) roughness. The physical mechanism involved during the IB of a bilayer is also explained.

## 2. Experimental

### 2.1. Samples and AFM Characterization

The samples discussed in this study are selected from our previous study [23,24]. The detailed experimental parameters of each sample, such as ion energy, incidence angle and fluence, are available in [23,24]. All of the samples were bombarded with Ar^+^ at an incidence angle of 50°. Fused silica substrates were used for coating. The ARC and PR were spin coated. For the Au/ARC sample, Au was deposited on the ARC by ion beam sputtering. The ion energy was 400 eV, and the ion current density was 240 µA/cm^2^ measured at normal incidence. The four samples correspond to the nanoripples on a single antireflection coating (ARC), a single photoresist (PR) layer, a PR/ARC bilayer, and a Au/ARC bilayer. The types of ARC and PR are AZ BARLi^®^Ⅱ200 [25] and AZ^®^ MiRTM 701 (14 cps) [26], respectively. The thicknesses of the single ARC and PR layers are ~430 nm and ~1500 nm, respectively. For the PR/ARC bilayer, the thicknesses of the PR and ARC are ~560 nm and ~430 nm, respectively. For the Au/ARC bilayer, the thicknesses of Au and ARC are ~200 nm and ~430 nm, respectively. During IB, a hot filament neutralizer was used to provide adequate conductivity of the surfaces and avoid the charging effect of the insulating layers. The ratio of the current of the hot filament neutralizer to that of the cathode filament was set as ~1.2–1.5. Appendix A show the main conditions of irradiation at different stages, which are added to the supplementary.

The surface morphology of all samples was measured with AFM. Operating in ScanAsyst mode, an atomic force microscope from Bruker Dimension ICON was used to characterize the surface topography of all samples. The model of the AFM tip used in this study is SCANASYST-AIR. The nominal radius and average string constant of silicon probes are 2 nm and 0.4 N/m. The pixel resolution of each AFM scan was at least 1024 × 1024 pixels. The wavelength (λ), roughness (σ), and power spectral density (PSD) of each AFM image were evaluated by using SPIP^TM^ software [27]. The PSDs were calculated along the direction of the ripple wave vector. Then, the PSD [28] and rms roughness [13] of each sample were extracted from the original AFM data. For convenience, the original AFM images and the corresponding PSDs and roughnesses of all samples are shown in this paper.

Note that the PSD is calculated by the Fourier transform of surface height [28] (i.e., the statistics of its morphology in frequency space). For a PSD curve, a coordinate of (*f*_i_, *p*_i_) corresponds to ripples with a spatial frequency of *f*_i_ and a probability of *p*_i_. The area of each AFM image is 5 × 5 µm^2^ with a resolution of 1024 × 1024 pixels. The spatial frequency of the PSD of each AFM image ranges from 0 to 0.1022 nm^−1^ with an interval of 0.0002 nm^−1^. There are 512 components of lateral frequencies for each PSD curve.

As shown in the PSD curves, the spatial frequency *f*_0_ of 0.005 nm^−1^ (corresponding to spatial wavelengths of 200 nm (i.e., the reciprocal of spatial frequency)) denoted by dashed lines in orange corresponds to a minimal intensity. The frequency range less than (more than) 0.005 nm^−1^ in the PSD curve is defined as the low-frequency (high-frequency) portion. Thus, the overall, low-frequency and high-frequency roughnesses of all samples are calculated [14]. The low- and high-frequency regions can also be called long- and short-wavelength regions.

In our experiments, at each irradiation condition, a 30 mm × 30 mm sample was used for bombardment. Three locations in or around the specimen center were measured. The error bars of the PSD entropy, rms roughness and wavelength of each image were calculated from the standard deviation of three measurements of one experimental run.

### 2.2. PSD Entropy

Shannon information entropy [29] is used to characterize the uncertainty of occurrence of events. In this study, PSD entropy [30], *H*, is used to characterize the degree of order of self-organized ripples. The definition of *H* is shown in Equation (1),
(1)$$H=-\sum_{i=1}^{N}p_{i}\log p_{i}$$

As illustrated in Section 2.1, in Equation (1), N represents the number of lateral frequencies, *p*_i_ represents the occurrence probability of the i^th^ spatial frequency, and the sum of the probability of nanoripples with all frequencies is 1. For patterns with only a certain spatial frequency, its PSD entropy has the minimum value *H* = 0. The initial “smooth” surface is considered a surface morphology with various uniformly distributed frequency components. Thus, its PSD entropy has the maximum value *H* = logN.

The function “Average X-Fourier PSD × 1” of the SPIP software was used to analyze these nanoripples. This indicates that the spatial frequency of the PSD curves in this study ranges from 0 to 0.1022 nm^−1^ with an interval of 0.0002 nm^−1^. The minimal frequency of the PSD should be 0.0002 nm^−1^, which corresponds to the scanning length of an AFM image of 5 μm. The frequency component of zero does not make sense and is excluded from the PSD curves. Hence, the N for the PSD entropy is decreased from 512 to 511. In principle, the maximum value of the PSD entropy for the analyzed system is log 511. 

Note that whether low- and high-frequency regions are divided or not, together with the value of *f*_0_, depends on the specific profiles of PSD curves of nanoripples.

## 3. Results and Discussions

### 3.1. General Temporal Evolution of Morphologies of the Surfaces

Figure 1a–e, Figure 2a–e, Figure 3a–e and Figure 4a–e show the temporal evolution of the AFM images of the surface morphology of the four samples (i.e., the single ARC layer, single PR layer, PR/ARC bilayer, and Au/ARC bilayer). The corresponding PSD curves of each AFM image are shown in Figure 1a’–e’, Figure 2a’–e’, Figure 3a’–e’ and Figure 4a’–e’. Appendix A show the PSD plots in logarithmic scales of the AFM images of Figure 1a–e, Figure 2a–e, Figure 3a–e and Figure 4a–e. The five AFM images of each sample display the corresponding morphologies at five typical stages (Stages A–E). For the two bilayer samples, Stage A corresponds to the well-formed PR and Au ripples, i.e., Figure 3a and Figure 4a. The images of Figure 3b–e and Figure 4b–e, corresponding to Stages B–E, were tested on the ARC of the PR/ARC and Au/ARC bilayer systems, respectively. As shown in the AFM images, for each sample, with increasing bombardment time, the self-organized nanoripples evolve from growth, by being well ordered, to degradation [23,24]. In general, the degree of ordering of the well-grown ripples on the bilayer systems (Figure 3d and Figure 4d) is better than that of the well-developed monolayers (Figure 1b and Figure 2c). Moreover, the degree of ordering of the ripples on the ARC surface is better than that on the PR surface. However, the longitudinal continuity of the ripples on the ARC surface is worse than that on the PR surface. Note that PR corrugated ridges are bent with a bean-like structure. The ARC ridges are short. In our previous study [23,24], the PSD curves and autocorrelation lengths of AFM images were mainly used to characterize the degree of ordering of ripples and the continuity of longitudinal ripples, respectively. In this paper, PSD entropy is introduced to give a comprehensive characterization of the degree of ordering of ripples and compared with the PSD curve and roughness.

### 3.2. Comparison of the Parameters for the Temporal Evolution of Nanoripples between PSD Curves and PSD Entropy

According to Equation (1), the PSD entropy is calculated from the PSD curves. To confirm that reliable statistics in the PSD entropy analysis can be obtained by using the AFM image with an area of 5 μm × 5 μm, we compare the PSD entropy calculated from different scanning lengths of AFM images (see Appendix A). The PSD entropy value is used to characterize the degree of ordering of the nanoripples. The lower the PSD entropy of the nanoripples is, the higher the degree of ordering of the nanoripples. The corresponding PSD curve shows the distribution of various spatial frequencies that may exist in the lateral direction of the nanoripples. First, for nanoripples, the PSD entropy of nanoripples with uniform height and the same lateral periodicity does not depend on the scanning length or size of grating patterns and absolute ripple height (Appendix A). The PSD entropy of gratings with different absolute ripple heights may shift up or down the amplitude of the evaluated PSD function, which can be deduced from the simulation data shown in Appendix A. Second, the PSD entropies are vulnerable to the height fluctuation of gratings with a specific average height, as shown in Appendix A; the larger the height fluctuations are, the higher the PSD entropy is. In fact, the amplitudes of the PSD curves of these gratings near the base frequency are almost identical [orange line in Appendix A]. The amplitudes of the PSD curves abruptly decrease at least four magnitudes less than that of the base frequency as the frequency deviates from the base frequency. Experimentally, the maximum value of the PSD entropy of a single ARC and PR layer system corresponds to that of the initial morphology of the randomly deposited (i.e., spin-coated ARC and PR layer (see horizontal dashed lines in Figure 5a–d)). For the single ARC and PR layers, and Au/ARC bilayer, with increasing bombardment time, the value of the PSD entropies decreases (Figure 5a,b,d), which corresponds to the emergence of high-frequency peaks in the PSD curves or an increased lateral periodicity of ripples. This also agrees well with the observation that the surface morphology gradually grows into a quasiperiodic grating structure.

According to the AFM morphology, the degree of ordering of the well-developed nanoripples (Figure 3d and Figure 4d) on the bilayer systems is better than that of the PR (Figure 2b) and ARC (Figure 1c) monolayers. However, the absolute value of PSD entropy of the bilayer systems is not significantly lower than that of the monolayers. Therefore, it is not suitable to evaluate the degree of ordering of nanoripples based on absolute values of PSD entropies. This can be attributed to the fact that the regularity of patterns is characterized by lateral and vertical regularity. The PSD and PSD entropy in this study only relates to the lateral regularity, i.e., the degree of ordering of patterns in this study.

The calculation of PSD entropy does not depend on the shape of the PSD curve. PSD entropy can be calculated for PSD curves even with an imperfect PSD peak. Appendix A shows the lateral correlation length of the nanoripples on ARC, PR, PR/ARC and Au/ARC surfaces. In general, for a nanorippled morphology, the value of the error bar of its PSD entropy is less than that of the corresponding lateral correlation length. 

### 3.3. Comparison of the Parameters for the Temporal Evolution of Nanoripples between Roughness and PSD Entropy

The temporal evolutions of the roughness and PSD entropy of each sample generally go by contrary. With increasing time, the nanoripples on the surface of each sample develop and become increasingly pronounced. Therefore, the overall rms roughness of all samples (Figure 6a–d) first increases and then becomes stable, which indicates the vertical growth of ripples, while the PSD entropy (Figure 5a–d) of all samples first decreases and then remains stable, which demonstrates the high degree of ordering of ripples. For the PR/ARC and Au/ARC bilayers, the wavelengths of the corresponding nanoripples evolve at the level of ~100 nm at stages C–D (Figure 7c,d). Note that, if there is no peak in a PSD curve (see Figure 4a’,b’), then neither a pattern wavelength nor the corresponding correlation length can be defined (see Figure 7d and Appendix A). In contrast, the ordering of a surface, even without remarkable patterns, can be characterized based on the PSD entropy. In particular, the PSD entropy analysis would be useful to quantify the weak order of saw-tooth patterns such as those reported in Figure 2b of ref [31].

In general, at Stages C–D, the lateral ordering of the nanoripples refers to the high-frequency portion of the PSD and high-frequency roughness. Correspondingly, the high-frequency range of a PSD curve depends on the pattern wavelength. With increasing ion dose (Stages D–E), surface roughening and vertical disorder become pronounced due to nonlinear terms of the equation of motion starting to operate [32]. Hence, the overall portion, particularly the low-frequency portion of the PSD curve and roughness need to be considered. They are demonstrated in detail as follows:

The temporal evolution of the nanoripples on bilayer surfaces is clearly shown in the curves of the roughness and PSD entropy of the high-frequency nanoripples (Figure 5c,d and Figure 6c,d). In contrast to those of a single PR or ARC, from stages B to D, the roughness of high-frequency ripples remains at a relatively stable and high level. Indeed, at these stages, the height of the nanoripples is uniform since latent nanoripples on the ARC surface can be induced by IB, which is verified in ref [23]. For comparison, the profile of the ARC nanoripples shown in Figure 8c_2_ is denoted as the dashed profile in Figure 8c_3_, which shows that the heights of the nanoripples may increase or decrease.

For all samples, with the development of nanoripples, the PSD entropy of high-frequency nanoripples continuously decreases. This indicates the improved degree of ordering of the nanoripples, which agrees well with the occurrence of high-frequency peaks in the corresponding PSD curves.

With the guidance of the nanoripples on the top layer, the degree of ordering of the nanoripples on the ARC surfaces of both PR/ARC and Au/ARC bilayers is significantly improved. The involved mechanisms of the whole process are explained in Section 3.4.

Note that, regarding the PR/ARC case, the evolutions of the overall and high-frequency roughnesses appear different. This can be attributed to not the high-frequency but the low-frequency portion dominant in the nanoripples in their PSD curves (Figure 3a’–e’). This also suggests that it is important to distinguish the influence of high- and low-frequency content distributions on the evolution of ripple characteristics in the process of characterizing the evolution of nanoripple morphology. Thus, special attention should be given to samples with large portions of low-frequency fluctuation.

In general, the degree of ordering of self-organized nanoripples refers to the high-frequency portion of the pattern. Although the temporal evolution of the overall roughness slightly decreases, the high-frequency roughness is relatively stable, indicating that the corrugated structures do not degrade under IB.

### 3.4. Scenario of the Guided Self-Organization Produced by Ion Bombardment of a Bilyaer System

Due to the complexity near the interface of the top and underlying layers during bombardment, it is still a major challenge for recent theoretical models to simulate the IB-induced nanoripples on a bilayer. Thus, we provide a scenario based on our observations [23,24] and the basic physical mechanisms of IB. Consistent with the proposed mechanism in Figure 5 in ref [23], Figure 8 shows the mechanisms during bombardment of a PR/ARC bilayer in detail.

PR including Ethyl-(s)-lactate and n-Butyl acetate and ARC including Ethyl-(s)-lactate and 1-methoxy-2-propanol are multi-component materials. Therefore, the preferential sputtering effect may occur during IB based on Sigmund’s theoretical study [33]. Moreover, according to our investigation on PR at normal incidence, IB-induced decomposition of PR can be even stronger than the preferential sputtering effect in PR [34,35].

Considering the PR/ARC bilayer, for instance, a possible scenario is shown in Figure 8. The proposed scenario involves two key processes, which rarely concurrently develop during IB. One is the pattern transfer using a sacrificial IB-induced nanorippled mask (Figure 8c_1_,c_2_), which is similar to a pattern transfer process from the PR mask (Figure 8a_1_) to the ARC pattern (Figure 8a_2_) during conventional lithography. The other is the subsequent curvature-dependent sputtering (Figure 8c_2_,c_3_) based on the Bradley-Harper theory [12] (i.e., the IB-induced nanoripples on the initially flat surface (Figure 8b)). It is unique and crucial that the profile of the prepatterned ARC ripples (dash profile in Figure 8c_3_ or solid profile in Figure 8c_2_) can be increased into that of the well-grown ripples (solid profile in Figure 8c_3_).

Note that there is no shadowing effect during IB. The height of ripples in Figure 8b_2_,c_2_,c_3_ may be larger than it actually is, to clearly demonstrate its increase and decrease at Stages C–D.

Furthermore, according to our experimental results in refs [23,24], first, the quality of underlying ARC ripples strongly depends on that of nanoripples on the top layer, corresponding to the nanoripples on the PR surface in Figure 8c_1_ or Figure 3a or Figure 4a. Hence, the thickness of the top layer is selected to ensure well-formed ripples on the PR surface. Second, it is essential that the intrinsic nanoripple period of the top layer material be as close to that of the underlying layer material as possible to realize the guidance of the nanoripples. In this regard, the PR nanoripples, as shown Figure 8c_1_ or Figure 3a or Figure 4a, act as an external stimulus to guide the growth of the intrinsic nanoripples on the underlying surface.

## 4. Conclusions

In conclusion, within the framework of guided self-organization, we have revisited the ion bombardment of a bilayer target system. Since the issues of guiding self-organization exist in multiple disciplines, the enhanced nanoripples produced by ion bombardment of bilayer systems may provide unique inspirations for other fields.

Combined with our previous findings on the nanoripples produced on PR/ARC [23] and Au/ARC [24], our experimental findings reveal that the self-organized nanoripples on the top target material can guide the growth of the self-organized nanoripples on the underlying target material if the experimental conditions are selected appropriately. The template with a self-organized nanostructure may alleviate the critical requirement for periodic templates with a small period of ~100 nm. These guided nanoripples on the underlying surfaces have a much higher degree of ordering than either of its single layers. This is a consequence of the sacrificial IB-induced nanorippled mask that generates prepatterned nanoripples on the underlying surface and of the resulting coupling between the prepatterned and latent intrinsic nanoripples due to curvature-dependent sputtering on the underlying layer. Ion beam-induced interface mixing in bilayers may be a driving force for guided assembly during the bombardment of bilayers. Thus, further study on the irradiated interface of bilayers is needed by using physical and chemical characterization and even simulation.

In this study, the power spectral density (PSD) entropy of PSD curves calculated from atomic force microscopy data is introduced to characterize the lateral periodicity of nanoripples. The temporal evolutions of PSD entropy and rms roughness of nanoripples generally go by contrary. The contributions of high-frequency and low-frequency nanoripples to the characteristic parameters of nanoripples are clarified. In particular, comparisons of the characteristic parameters (i.e., roughness and PSD curves) of the high-frequency nanoripples between the bilayer and single-layer systems show that the quality of the nanoripples on the bilayer systems is remarkably improved.

In the future, we will calculate the entropies of the structural parameters of nanopatterns (e.g., the height and width of nanoripples). Moreover, we will develop other parameters based on information entropy (e.g., to characterize the longitudinal and holistic degree of ordering of nanoripples).

## Figures and Tables

**Figure 1 entropy-25-00170-f001:**
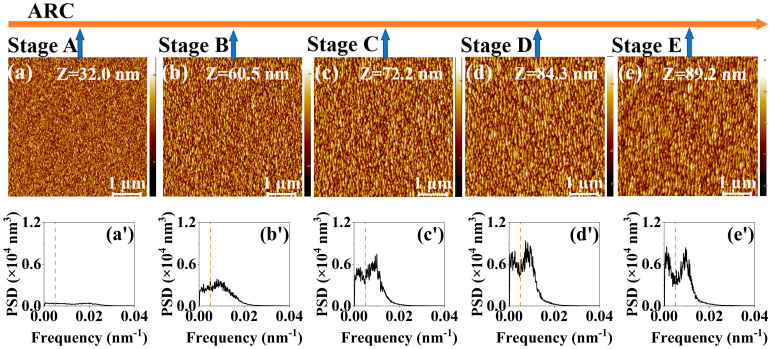
(**a**–**e**) AFM images of the irradiated ARC surface at 400 eV ion energy and ion-incidence angle of 50° for bombardment stages A–E. The height scale Z is specified in each image. (**a’**–**e’**) PSD curves of the AFM images shown in Figure 1a–e.

**Figure 2 entropy-25-00170-f002:**
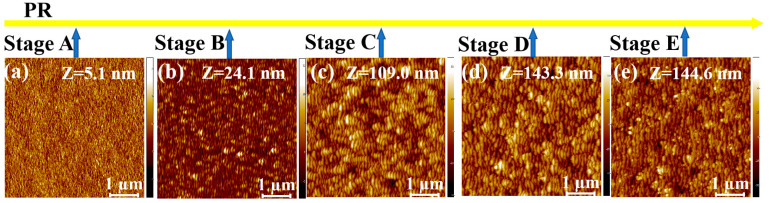
(**a**–**e**) AFM images of the irradiated PR surface at 400 eV ion energy and ion-incidence angle of 50° for bombardment stages A–E. The height scale Z is specified in each image. (**a’**–**e’**) PSD curves of the AFM images shown in Figure 2a–e.

**Figure 3 entropy-25-00170-f003:**
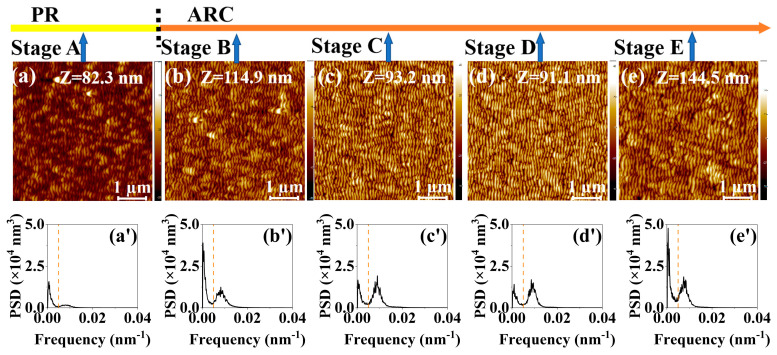
(**a**–**e**) AFM images of the irradiated PR/ARC surface at 400 eV ion energy and ion-incidence angle of 50° for bombardment stages A–E. The height scale Z is specified in each image. (**a’**–**e’**) PSD curves of the AFM images shown in Figure 3a–e.

**Figure 4 entropy-25-00170-f004:**
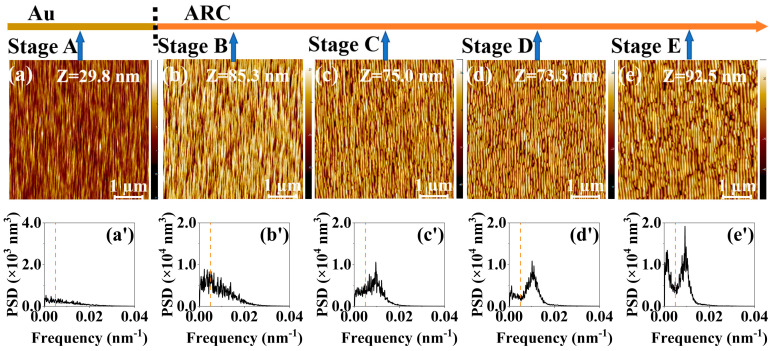
(**a**–**e**) AFM images of the irradiated Au/ARC surface at 400 eV ion energy and ion-incidence angle of 50° for bombardment stages A–E. The height scale Z is specified in each image. (**a’**–**e’**) PSD curves of the AFM images shown in Figure 4a–e. Figure 4a,b are cited from ref [24].

**Figure 5 entropy-25-00170-f005:**
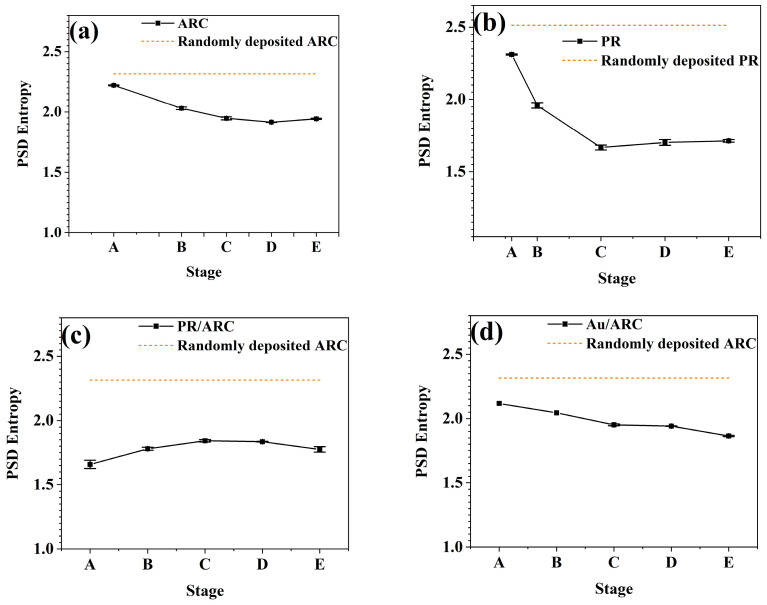
PSD entropy of the nanoripples on (**a**) ARC, (**b**) PR, (**c**) PR/ARC bilayer, and (**d**) Au/ARC surfaces from the corresponding AFM morphologies for bombardment stages A–E.

**Figure 6 entropy-25-00170-f006:**
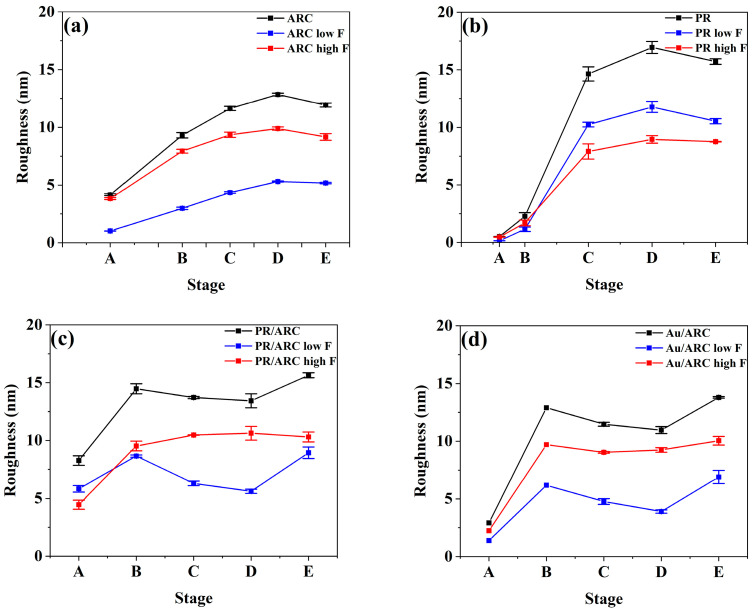
Overall roughness, high-frequency and low-frequency roughnesses of the nanoripples on (**a**) ARC, (**b**) PR, (**c**) PR/ARC bilayer [23], and (**d**) Au/ARC [24] surfaces from the corresponding AFM morphologies for bombardment stages A–E.

**Figure 7 entropy-25-00170-f007:**
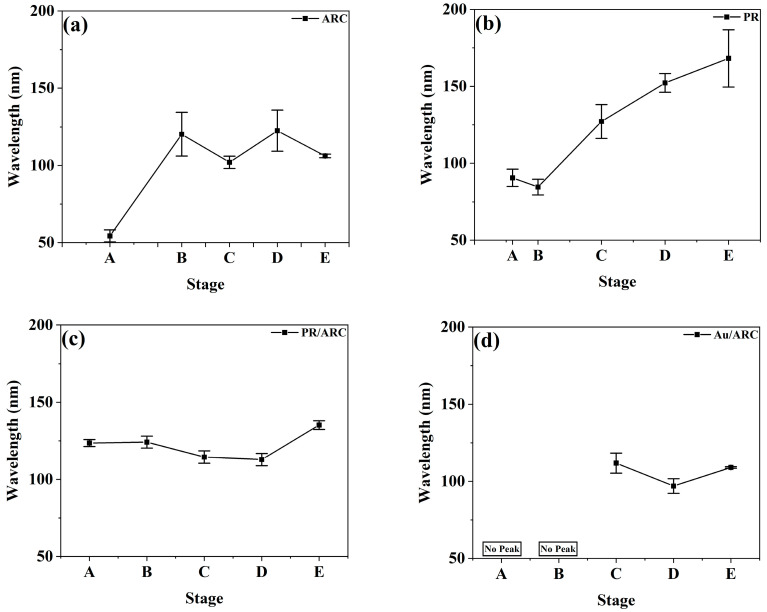
Evolution of the ripple wavelength of (**a**) ARC, (**b**) PR, (**c**) PR/ARC bilayer, and (**d**) Au/ARC bilayer.

**Figure 8 entropy-25-00170-f008:**
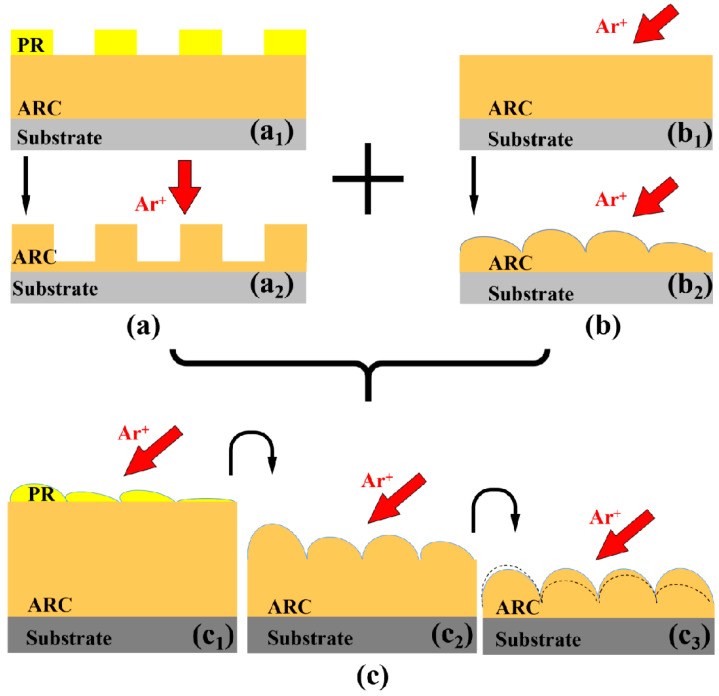
Schematic of (**a**) pattern transfer using a grating mask during conventional lithography, (**b**) curvature-dependent sputtering based on the Bradley-Harper theory [12], and (**c**) guided self-organization during ion bombardment of a bilayer system (i.e., the synergy of (**a**,**b**)).

## Data Availability

The data that support the findings of this study are available from the corresponding author upon reasonable request.

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
