# Peer review of "Can One Series of Self-Organized Nanoripples Guide Another Series of Self-Organized Nanoripples during Ion Bombardment: From the Perspective of Power Spectral Density Entropy?"

_entropy, 2023, doi:10.3390/e25010170_

Round 1
Reviewer 1 Report
This manuscript by Li et al presents a revisited study of the morphological evolution of self-organized nanoriples induced by ion bombardment, using an entropy parameter to analyze AFM images. Conceptually, it’s interesting to develop a parameter that can describe the ordering process in a concise yet precise manner, to uncover the underpinning principles from the complex self-organization phenomena. The parameter introduced in this work, the power spectral density (PSD) entropy- a simple form of Shannon information entropy, should be improved so that it can accurately and quantitatively describe the ordering and provide new insights into the self-organization process. For example, as stated by the authors, this entropy is calculated from PSD, which only captures the lateral direction of the ordering, so it cannot yield any additional information beyond PSD can be provided. Even worse, such entropy, with reduced information compared with PSD, is unable to discern the remarkable differences between nanoripples that form on single or bilayer surfaces. Thus, I strongly recommend that the authors derive the entropy parameters directly from the AFM images, which could provide new insights into the ordering. Overall, the manuscript exhibits much lower quality than the average of this group's previous publications. In my opinion, this work needs to be highly improved before it can be accepted by Entropy.
Some detailed comments/questions,
-
This work contains seven figures, but five (or more) of them are copied from their previous publication. I understand that the primary goal of this work is to re-analyze the previously published AFM data, but please consider a better way to present the published data rather than copy and paste? Also, the authors used different terms, including time, regimes, and stages, in figures 1-4 to label the AFM images. It’s very confusing without providing detailed description for such labels. Please consider using a unified label.
-
Some statements do not agree with the data. For example, line 149: “For all samples, with increasing bombardment time, the value of the overall PSD entropies decreases (Figures 5(a–d)),”. Actually, 5(c) shows an increase of the PSD entropy. What’s the error bar for the entropy?
-
The proposed mechanism (Fig7) seems inconsistent with their published paper, Fig 5, Nanotech 2021. It is unclear why the authors propose a different self-organization process since the paper uses no new data and data analysis does not provide any new insight.
-
Again, I strongly recommend that the authors derive the entropy parameters directly from the AFM images. I wonder whether such parameter can show a transition (e.g., a sudden change) around the regime (B) shown in Nanotech 2021, namely, the beginning of the irradiation of the second layer.
Reviewer 2 Report
The paper by Li et al. is interesting since it proposes a new approach to parametrize the order in IBS ripple induced patterns. In this sense, it is a new contribution in the field that deserves to be considered. However, in order the paper can be published many points have to be addressed and the manuscript improved in order the reader can know the limits and potential of the method. In the following, I detail these points.
- In the introduction, the authors state that “a concise parameter is not available to characterize the degree of ordering of nanostructures “. This is not true as can be seen in B. Ziberi et al. Phys. Rev. B 72, 235310 (2005) and R. Gago et al. Appl. Phys. Lett. 89, 233101 (2006). In the former paper is stated “the system correlation length is a measure for the lateral ordering of structures on the surface. It is calculated from the full width at half maximum FWHM of the first-order PSD peak and is inversely proportional to the FWHM”. Thus, this method is based also on the PSD functions of the irradiated surfaces. This has to be mentioned. Furthermore, in order to test the entropy method proposed here, the results obtained with the entropy method on the measured surfaces have to be compared with those obtained with the method proposed by Ziberi et al. on the same surfaces. I want to note that one of the authors of Ziberi et al. is also author of the present manuscript.
- One aspect that I do not like of the manuscript is the absence of details of the irradiated samples and some of the conditions of the irradiations. These data are skipped by referring to previous work. Even in this case, when referring to PR and ARC layers, their composition should be provided. This is important because when a compound material is irradiated, preferential sputtering effects can affect the pattern evolution. In my opinion, the experimental data have to be given in the manuscript. Also, details about the AFM employed and the tips used are missing. Likewise, it should be explained whether the PSDs are obtained using a homemade software or a commercial one. Other relevant point is that the authors seem to calculate the PSDs along the ripple wave vector direction, as the y-axis appears in length to the third exponent. This fact has to be clarified.
- The explanation in lines 97-101 is unclear and not satisfactory. For me, it is not clear whether the authors use AFM images of 4096 pixels for the PSD analysis or they interpolate the data obtained from 512 x 512 AFM images. In addition, the statement that “the spatial frequency of the PSD of each AFM image ranges from 0 to 0.1024 nm-1” is not correct since frequency 0 means infinite length.
- The criterion used in lines 102-104 to define the low-frequency and high-frequency regions seems arbitrary. The authors should explain it better. Moreover, if the authors propose this new method to assess the ripple ordering, they should provide with a more general criterion based on the value of the ripple wavelength. In particular, I guess that it would better to define these values from the value of the wavelength. In this case, all the images have the same size but different systems will have different wavelengths, so the reader needs a criterion based on which image size is convenient to perform this analysis depending on the experimental ripple wavelength.
- The detailed calculation of the N values (4088, 193, and 3895) in line 117 has to be given. In addition, the last sentence in lines 118 and 119 has to be explained since, as it is, is rather obscure.
- Figures 1-3 are the SAME than those shown in reference 23. I let the editor to judge this aspect. Although it is true that reference 23 is an open access article, it would seem that the authors only have these images (data). New data would be welcome.
- Regarding the PSD plots, it would be useful to plot them also in logarithmic scales (this can be shown in supplemental information).
- I understand that it is difficult but it is somehow striking the absence of error bars in the entropy plots. Some estimation should be provided. Likewise, the error bars in some roughness graphs are absent.
- Graphs with the evolution of the ripple wavelength should be included.
- In section 3.3, I find the discussion a bit ambiguous. It is not clear whether just the high frequency region should be used for evaluation of the ripple ordering or the whole PSD entropy. A clearer conclusion is required. I guess that the high frequency part relates with the lateral ripple ordering whereas the whole PSD entropy also samples the vertical ripple modulation. It should be noted that with increasing ion dose the ripples become asymmetric and non-linear terms start to operate. These effects lead to surface roughening (as the authors state) and to the vertical disorder of the ripples. In this sense, perhaps it is better to refer just to lateral ordering of the ripples when using the high frequency portion of the PSD. At this point, I again stress the need to specify how to choose the high frequency window depending on the pattern wavelength. The authors should give an input to the reader to make this choice for her/his eventual system.
- Regarding section 3.4, I find the explanation of the scenario rather simplified. I guess (I do not know for sure since the authors do not give this information) that PR and ARC are not mono-elemental materials. Thus, preferential sputtering (i.e., compositional) effects can come into play affecting the patterning process. Thus, not only the BH model has to be invoked here.
- Figure 7 is somehow misleading. In schemes b and c, it seems that some parts of the ripples are hidden from the incoming ions, that is, there are shadowing effects. I guess this is not the case, but, in any case, the authors should check from the morphological analysis of the images whether this is the case, and change figure 7 accordingly.
Reviewer 3 Report
The manuscript entitled “Can one series of self-organized nanoripples guide another series of self-organized 1 nanoripples during ion bombardment: from the perspective of power spectal density entropy?” presents new analysis of the degree of ordering of nanostructures obtained by ion beam irradiation. The synthesis and characterization of the samples were presented in previous works. In this work power spectral density (PSD) entropy is introduced to evaluate the degree of ordering. The analysis extends to several types of samples (layers of photoresist, antireflection coating, Au and sequences of both) and different times of ion irradiation.The authors use the analysis of PSD entropy to conclude that the ion beam-induced nanoripples on the top layer may act as a self-organized template to guide the development of latent nanoripples on the next below layer.
The results are well described and discussion is rationally presented and endorse the conclusion.
Regarding the English language and style, I am not English native but if there are mistakes, they are minor.
My concern is that although the samples were selected from already published works, all the experimental parameters must be included in this manuscript. These descriptions must include a full description of the materials e.g. which type of photoresist (commercial?), the antireflection layer. In addition, a description of the ion implanter, the AFM and the employed software for the PSD must be included.
In this sense, I would like to make a question: Can the authors’ methodology be implemented on other AFM images. i.e. there is an available commercial software or is author’s proprietary.
If the authors include the full experimental details and reply my question I would recommend its publication.
Reviewer 4 Report
This paper deals with periodic surface pattern formation induced by ion bombardment and guided assembly for bilayers, that is, interesting and hot topic for the development of high-quality nanofabrication processes. The authors introduce a new quantity, the PSD, to characterize the quality of the nanopatterns and to compare nanopatterns generated on mono- and bilayers. The paper may be published in the journal Entropy, however, the authors may give some more information on the used samples and the ion irradiation process, and some more discussion on the ion bombardment related effects and their possible role in pattern formation, see my comments below.
-In title: „spectal density” can be corrected as „spectral density”
-The curvature radius and string constant of the AFM tip as well as details on AFM scan measurement can be given.
-Details about type/composition, thickness, and resistivity features of the ARC and PR layer can be given.
-What is the penetration depth of the bombarding ions in the monolayer and bilayer structures?
-What should be the effect of ion beam induced interface mixing (if it occurs) in bilayers at the applied ion fluences? Should it be also a driving force for guided assembly?
-For insulaing layers like PR, what should be the effect of charging of the sample surface on nanopatterning, especially on the stability of pattern formation at the applied very low ion energy? The authors may comment on surface charge induced ion deceleratiom/acceleration, and ion deflection processes in the text.
-Have the authors applied any neutralization tool – e.g. electron shower – to reduce surface charging during ion bombardment of poorly conducting surfaces?
-What should be the effect of surface sputtering on the height of the nanopatterns and ripple widths at the applied ion fluences for the case of monolayers and bilayers? Should the effect of sputtering significantly differ for Au, ARC, and PR materials?
-For the AFM topology figures, the applied ion fluences can also be given for stages A-E.
-The authors may give comment in the paper: how strong is the correlation between the amplitude of the evaluated PSD function and the inhomogeneity of ripple height? I mean, for two nanopatterns with same lateral periodicity, how their different ripple height fluctuation (measured along perpendicular direction to the direction of lateral periodicity) will appear in their PSD curves?
Round 2
Reviewer 2 Report
The authors have made an evident effort to improve the manuscript by addressing my criticisms and suggestions. Although the manuscript is clearer now, I think that there are some issues that have to be corrected before it can be published.
My main concern refers to the data manipulation to obtain the PSD entropy. As the authors explain now in their response “for the PSD curves, the function “Average X-Fourier PSD×8” of the SPIP software was used to analyse these nanoripples with an eight-time higher frequency resolution [27] than that of the normal “Average X-Fourier PSD×1”.
Looking at the webpage quoted in [27] to know what implies this procedure, one finds the explanation that “to achieve the highest accuracy it will be an advantage to apply the FFT High Resolution function (Requires the Extended Fourier or Calibration Module) , which means that the profile will be mean value padded so that it contains 16 times more elements before calculating its Fourier transform.” This statement implies that the SPIP software for this purpose in fact interpolates the raw PSD data in order to increase the data sampling, and thus, to increase the points in the corresponding FFT. I find this incorrect. The authors should work with the raw data (1024 pixels) and repeat their analysis. For me, it is not meaningful to interpolate the data in order to have a narrower sampling in the FFT. Therefore, I find that in order the paper can be published, only analysis of the raw data should be presented.
This fact has relevant consequences since now N will be 512 (8 times smaller than 4096). Accordingly, the number of points in the so-called low-frequency region will be about 25. This very low figure prevents any reliable quantification and analysis. Consequently, the distinction made between the low- and high-frequency regions will disappear and the authors should work just with the PSD of the whole image. Accordingly, the authors should re-calculate the errors involved. Likewise, they will have to revise, if necessary, the advantages of the PSD entropy method compared to that of the correlation length stated in the SI.
Other points worthy to address are:
- I find the plots in figure S7 a-c misleading. The y-axis scale is so large that the change of the correlation length is not visually appealing. I guess that by setting the maximum value in a-b at 500 and in c at 1000, the evolution will be better shown. In addition, it should be clarified that the correlation length is proportional to the inverse of the PSD entropy in order these data can be better compared.
- Along the manuscript and the authors’ response it is stated that “PSD entropy can be calculated for PSD curves without an imperfect PSD peak or even no clear peak, e.g., the PSD curves of the morphologies at stages A and B of the Au/ARC sample” (I guess that instead of without the authors mean with). The authors should acknowledge that if there is no peak, then a pattern wavelength cannot be defined (see figure S7d). Then we are not dealing with the calculus of the order or entropy of a surface pattern but rather with that of just a surface. However, the PSD entropy analysis would be useful to quantify the weak order of saw-tooth patterns such as those reported in Fig 2b of Applied Surface Science 580 (2022) 152267. This figure also is useful to comment on other of the issues I commented in my former review that the authors have not addressed: in this figure, as bombardment proceeds, there is an evident coarsening, the wavelength increases, and the PSDs evolve towards low k regions. This is the reason why I asked initially: how large has to be the AFM image with respect to the wavelength in order to have a reliable statistics in the PSD entropy analysis? This sort of analysis, in the line of that presented in figures S5 and S6, will be very helpful for the reader.
- Connected to the previous comment, the authors should present the PSD entropy of an uncorrelated surface, such as that of a Si polished wafer or a simulated surface formed by random deposition. This, I guess, will give a maximum experimental value of the PSD entropy. In this line, and connected with the plots in figure 5, I think that it is necessary to indicate (for instance with horizontal dashed lines) the maximum values of the PSD entropy for the analyzed system. That is, in the present system log4088 (or log3895) and log193 for the high- and low-frequency regions. Note that in the revised version just the raw data have to be used, which implies that N = 512 and the differentiation between these two frequency regions will disappear.
